# Comparative analysis of potential drug-drug interactions in a public and private hospital among chronic kidney disease patients in Khyber Pakhtunkhwa: A retrospective cross-sectional study

**Roheena Zafar[1,2], Inayat Ur Rehman[1]\*, Yasar Shah[1], Long Chiau Ming[3], Hui Poh Goh[4]\*, Khang Wen Goh[5]**

1 Department of Pharmacy, Garden Campus, Abdul Wali Khan University Mardan, Mardan, Pakistan,
2 Department of Pharmacy, North West General Hospital and Research Center, Hayatabad Peshawar,
Pakistan, 3 School of Medical and Life Sciences, Sunway University, Bandar Sunway, Malaysia, 4 PAPRSB
Institute of Health Sciences, Universiti Brunei Darussalam, Bandar Seri Begawan, Brunei Darussalam,
5 Faculty of Data Science and Information Technology, INTI International University, Nilai, Malaysia

\* Inayat.rehman@awkum.edu.pk (IUR); pohhui.goh@ubd.edu.bn (HPG)

University, MALAYSIA

**Data Availability Statement:** All relevant data are
within the paper. Public access to all raw data is
restricted due to the consent from that participants

## Abstract

### Introduction

Chronic kidney disease (CKD) is a significant public health challenge due to its rising incidence, mortality, and morbidity. Patients with kidney diseases often suffer from various comorbid conditions, making them susceptible to potential drug-drug interactions (pDDIs) due to polypharmacy and multiple prescribers. Inappropriate prescriptions for CKD patients and their consequences in the form of pDDIs are a major challenge in Pakistan.

### Aim

This study aimed to compare the incidence and associated risk factors of pDDIs among a public and private sector hospital in Khyber Pakhtunkhwa, Pakistan.

### Method

A retrospective cross-sectional study design was conducted to compare pDDIs among public and private sector hospitals from January 2023 to February 2023. Patients profile data for the full year starting from January 1 2022 to December 302022, was accessed All adult patients aged 18 years and above, of both genders, who currently have or have previously been diagnosed with end-stage renal disease (ESRD) were included. For assessing pDDIs, patient data was retrieved and checked using Lexicomp UpToDate® for severity and documentation of potential drug-drug interactions.

agreed to. All data related queries can be addressed by Dr. Inayat Ur Rehman (inayat.rehman@awkum.edu.pk); and Mr. Shah Faisal (faisal@nwgh.pk).

**Funding:** The authors received no specific funding for this work.

**Competing interests:** The authors have declared that no competing interests exist.

## Results

A total of 358 patients' data was retrieved (with n = 179 in each hospital); however, due to incomplete data, n = 4 patients were excluded from the final analysis. The prevalence of pDDIs was found to be significantly higher in private hospitals (84.7%) than in public hospitals (26.6%), with a p-value <0.001. Patients in the age category of 41–60 years (AOR = 6.2; p = 0.008) and those prescribed a higher number of drugs (AOR = 1.2; p = 0.027) were independently associated with pDDIs in private hospitals, while the higher number of prescribed drugs (AOR = 2.9; p = <0.001) was an independent risk factor for pDDIs in public hospitals. The majority of pDDIs (79.0%) were of moderate severity, and a significant number of patients (15.1%) also experienced major pDDIs, with a p-value <0.001. The majority of pDDIs had fair documentation for reliability rating in both public and private hospitals.

## Conclusion

The prevalence of pDDIs was higher among CKD patients at private hospitals, and most of the pDDIs were of moderate severity. A considerable number of patients also experienced major pDDIs. The risk of experiencing pDDIs was found to be higher in older patients and among those prescribed a higher number of drugs.

## Introduction

Chronic kidney disease (CKD), due to its increased cases and morbidity and mortality, is considered a challenging global health problem [1, 2]. According to the 2019 study of the Global Burden of Disease (GBD), approximately 697 million CKD cases were reported worldwide [2]. In 2019, CKD was ranked as the eleventh leading cause of mortality and morbidity globally, resulting in 1.43 million deaths. Given the rise in CKD cases and mortality, it is expected that the number of cases will reach 4.0 million by 2040 [3, 4]. Furthermore, CKD patients experience worse clinical outcomes and compromised quality of life [5–8]. These patients with CKD often suffer from complications [9–11], i.e. diabetes mellitus [10, 12], cardiovascular disease (CVD) [13, 14] and hypertension [15, 16]; therefore, polypharmacy is inevitable and highly prevalent among these patients. The use of multiple medications for managing comorbidities further exacerbates the progression of CKD [10]. Polypharmacy in CKD is associated with increased healthcare costs, poor medication adherence, and significantly contributes to drug-related problems, including adverse drug reactions and drug-drug interactions (DDIs) [17, 18].

The consequences of DDIs can be life-threatening and may even lead to lethal toxicities [15]. Additionally, the pharmacokinetic and pharmacodynamic profiles of the majority of drugs excreted by the kidneys are altered as a result of CKD itself, which can contribute to the occurrence of DDIs [19]. DDIs result in an augmented risk of morbidity and mortality among CKD patients, diminished quality of life, and also prolonged hospitalization [20]. The estimated incidence of potential DDIs (pDDIs) varies from 3–5% among patients consuming fewer medicines, while among those who receive 10–20 medications, the chances increase to 20% [15]. The potential risks for pDDIs in CKD patients include increased patient age and an increase in the number of drugs [21]. The risk of pDDIs among CKD patients also increases to a greater extent when numerous prescribers are involved in the treating same patients while prescribing additional drugs for management [22]. The pDDIs are preventable and their early "identification and detection" are very crucial to undertake appropriate and adequate

preventive measures and interventions at an early stage [21]. As pDDIs are preventable, 46% of the hospital admissions resulting from pDDIs can be prevented [23], which could provide significant relief to the healthcare system [24].

In Pakistan, most public and private sector hospitals are funded by a government initiative named the Sehat Insaaf card, through which medication and medical care are provided free of charge to patients. However, inappropriate prescribing and the occurrence of pDDIs remained significant challenges for the healthcare system in Pakistan [25]. The multiple prescriptions by numerous physicians in Pakistan aggravate the disease progression. Therefore; this study aimed to compare the incidence of pDDIs among a public and private sector hospital of Khyber Pakhtunkhwa, Pakistan, in order to gain insight into these drug-related problems and design a national policy for practicing nephrologists in both public and private sector hospitals of Pakistan.

## Methodology

### Study design

The study was conducted at the nephrology units of two hospitals: Institute of Kidney Diseases, Peshawar Pakistan (a Public sector hospital), and North West General Hospital & Research Center, Peshawar Pakistan (a private sector hospital), using a retrospective cross-sectional study design. Clinical pharmacy services were not present at ward level in both hospitals, and screening of pDDIs via software-based was deficient. For research purpose, data were accessed for the full year starting from January 1 2022 to December 30 2022, from hospital systems/profiles. The eligible patient's profiles was collected within two months i.e. January 12023 to February 28 2023, based on the inclusion/exclusion criteria.

### Inclusion/Exclusion criteria

All adult patients of 18 years and above, of both genders, who currently have or have previously been diagnosed with end-stage renal disease were included. Patients' profiles lacking relevant data required for the study were excluded.

### Data source

Data of the CKD patients admitted to the nephrology units of both hospitals were extracted from their medical records. Patients profile including age, gender, length of stay in the hospital, CKD stage, serum creatinine, potassium level, blood urea nitrogen, number of drugs prescribed, generic names of drug prescribed, presence of comorbidities such as diabetes, hypertension, cardiovascular disease, hepatitis B and hepatitis C and other comorbidities, were recorded from the medical profiles/records of CKD patients.

### Screening for pDDIs

The evaluation of pDDIs was carried out with the help of Lexicomp®, which classified them based on interaction risk rating, severity, and reliability rating. The performance of Lexi-interact as a drug-drug interaction screening tool has been evaluated in multiple studies in the past [26, 27], and it is widely considered to be one of the most effective ones available. These studies have found that Lexi-interact is highly sensitive (87–100%) and specific (80–90%) in most cases [28–30]. The pDDIs were then categorized for severity and reliability rating. The severity rating is the reported or possible magnitude of interaction outcome, it is classified as **Minor** (minimal effects that are typically tolerable), **Moderate** (potential for significant interaction but not reaching the criteria for major severity), **Major** (potential for serious interaction that

typically demands medical intervention) and **contraindicated** (referring to drugs that must never be used together due to severe and life-threatening interactions) [31]. While, the reliability rating assesses the quantity and quality of documentation available for an interaction, and it is categorized as **excellent**, **good**, and **fair** [32].

## Sample size

A total of n = 358 patients were included in the study, with 179 patients from each hospital. The sample size was determined based on the anticipated incidence of CKD (12.5%) [33] and calculated using a recommended formula [34] with a confidence interval of 95% and a precision of 5%.

## Ethics approval

The study was approved by the ethics committee of Abdul Wali Khan University Mardan (Approval no: EC/AWKUM/2021/27), and the Institutional Review Board of North West General Hospital & Research Center (Approval no: NWGH/DMER/EC/1726) and the Institute of Kidney Diseases, Peshawar, Pakistan (Approval no: 454). As it is a retrospective study, all data was fully anonymized before being accessed, and the Institutional Review Board of North West General Hospital & Research Center, as well as ethics committee of the Institute of Kidney Diseases, waived the requirement for informed consent.

## Statistical analysis

Data analysis was performed using SPSS version 22.0®. Descriptive statistics were used to present demographic characteristics in terms of frequencies and percentages. An independent t-test was employed to assess the difference between both hospitals. A multivariate binary logistic regression was also performed to identify the association of various predictors and risk factors with all pDDIs. Before multivariate binary logistic regression, a univariate logistic regression was performed and those factors having p-value <0.25, subjected to multivariate logistic regression. The findings of the logistic regression were expressed in odd ratio (OR) and 95% confidence intervals while the p-value <0.05 was considered as statistically significant.

## Results

A total of 358 patients were included in the study, with 179 from each hospital. Four patients (2 in each hospital) had incomplete data and were excluded from the final analysis. In the public hospital majority of patients were male (74%), while in the private hospital, 58.8% were male. The highest percentage of patients in the public hospital (46.9%) were in the age group of 41–60 years, whereas in the private hospital, 48.0% were in the age group of more than 60 years. The maximum hospital stay in the public hospital was higher compared to the private hospital, with 57.1% staying for 3–4 days, while in the private hospital, 40.1% stayed for less than 2 days. Most patients in the public hospital (65.0%) were prescribed two drugs, while in the private hospital, all the patients were prescribed more than five drugs. The demographic characteristics of patients in both public and private hospital were statistically different as shown in Table 1.

Table 2 presents the severity and documentation of pDDIs. When comparing the number of pDDIs between the two hospitals, the private hospital had a higher number of pDDIs compared to the public hospital. In terms of severity level, 27.2% of pDDIs were categorized as moderate in the public hospital compared to 79.0% in the private hospital. Furthermore,

**Table 1. Demographic characteristics of patients in both hospitals (n = 354).**

| Variables | Hospital | | | | P-value |
|---|---|---|---|---|---|
| | Public Hospital (n = 177) | | Private Hospital (n = 177) | | |
| | N | % | N | % | |
| **Gender** | | | | | |
| Female | 46 | 26 | 73 | 41.2 | **0.002*ᵃ** |
| Male | 131 | 74 | 104 | 58.8 | |
| **Age (Years)** | | | | | |
| = <40 | 68 | 38.4 | 23 | 13 | **<0.001*ᵃ** |
| 41–60 | 83 | 46.9 | 69 | 39 | |
| >60 | 26 | 14.7 | 85 | 48 | |
| **Hospital Stay** | | | | | |
| = <2 | 30 | 16.9 | 71 | 40.1 | **0.002*ᵃ** |
| 3–4 | 101 | 57.1 | 61 | 34.5 | |
| >4 | 46 | 26 | 45 | 25.4 | |
| **No Prescribed Drugs** | | | | | |
| 2 | 115 | 65 | 0 | 0 | **<0.001*ᵃ** |
| 3–4 | 58 | 32.8 | 0 | 0 | |
| = >5 | 4 | 2.3 | 177 | 100 | |
| **Comorbidities** | | | | | |
| No | 42 | 23.7 | 61 | 34.5 | **0.026*ᵃ** |
| Yes | 135 | 76.3 | 116 | 65.5 | |
| **No of Comorbidities** | | | | | |
| 0 | 42 | 23.7 | 61 | 34.5 | **0.005*ᵃ** |
| 1 | 79 | 44.6 | 80 | 45.2 | |
| = >2 | 56 | 31.6 | 36 | 20.3 | |
| **Comorbidities **** | | | | | |
| Hypertension | 106 | 59.9 | 24 | 13.6 | - |
| Diabetes Mellitus | 45 | 25.4 | 20 | 11.3 | |
| Hepatitis C Virus | 18 | 10.2 | 38 | 21.5 | |
| Heart Disease | 16 | 9 | 14 | 7.9 | |
| Urinary Tract Infection | 14 | 7.9 | 6 | 3.4 | |
| Hepatitis B Virus | 7 | 4 | 64 | 36.2 | |
| Benign Prostate Hyperplasia | 2 | 1.1 | 9 | 5.1 | |
| Asthma | 0 | 0 | 9 | 5.1 | |
| COPD | 0 | 0 | 1 | 0.6 | |
| Parkinson | 0 | 0 | 1 | 0.6 | |
| Tuberculosis | 0 | 0 | 2 | 1.1 | |

a: Chi-square test was performed

*p < 0.05 statistically significant

**(Diabetes mellitus, hypertension, hepatitis C) were the most common comorbidities observed in patients. Figures were > 100% as patients may be suffering from more than one chronic condition

25.5% of pDDIs were categorized as fairly documented in the public hospital, while 72.0% of pDDIs in the private hospital fell under this category [details are shown in Table 2].

Regarding the incidence of pDDIs, as shown in Fig 1, the incidence of pDDIs were significantly higher among patients in private hospital i.e., 84.7% as compared to 26.6% in public hospital having p-value <0.001.

**Table 2. Severity and documentation levels of pDDIs.**

| Variables | Public hospital (n = 177) | | Private hospital (n = 177) | | P-value |
|---|---|---|---|---|---|
| | n | % | n | % | |
| **Number of pDDIs** | | | | | |
| 0 | 129 | 72.9 | 27 | 15.3 | **<0.001**[*] |
| 1 | 43 | 24.3 | 20 | 11.3 | |
| 2 | 4 | 2.3 | 24 | 13.6 | |
| 3 | 0 | 0 | 20 | 11.3 | |
| 4 | 0 | 0 | 13 | 7.3 | |
| 5 | 0 | 0 | 12 | 6.8 | |
| >5 | 0 | 0 | 61 | 34.5 | |
| **Severity Levels** | | | | | |
| Major | 1 | 0.6 | 154 | 15.1 | **<0.001**[*] |
| Minor | 1 | 0.6 | 33 | 3.2 | |
| Moderate | 49 | 27.2 | 806 | 79.0 | |
| **Documentation** | | | | | |
| Excellent | 0 | 0 | 31 | 3.0 | **<0.001**[*] |
| Fair | 46 | 25.5 | 734 | 72.0 | |
| Good | 5 | 2.8 | 210 | 20.6 | |
| Poor | 0 | 0 | 18 | 1.8 | |

Chi-square test was performed; [*] p < 0.05 statistically significant

The comparison of private and public hospitals based on patient variables is presented in Table 3. All variables were significantly different in both hospitals except for hospital stay (p = 0.519). Age (p = <0.001), number of drugs (p = <0.001), and number of drug interactions (p = <0.001) were significantly higher in the private hospital compared to the public hospital. On the other hand, the number of comorbidities (p = 0.013) and the stage of CKD (p = <0.001) were significantly higher in the public hospital compared to the private hospital.

Regarding the multivariate regression model, the age category of 41–60 years (AOR = 6.2; p = 0.008), and the higher number of prescribed drugs (AOR = 1.2; p = 0.027), were

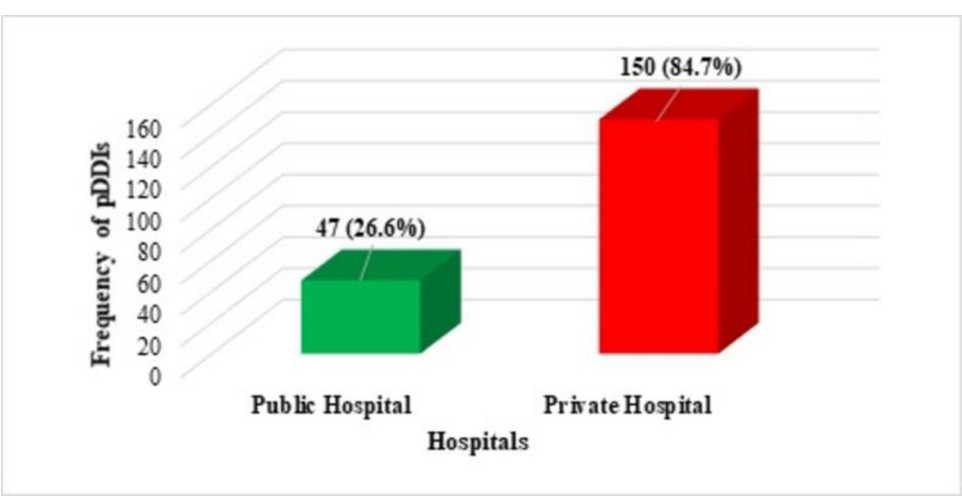

**Fig 1. Incidence of pDDIs hospital wise among selected patients.**

**Table 3. Comparative analysis of private hospital and public hospital.**

| Variables | Private Sector Hospital | Public Sector Hospital | p-value |
|---|---|---|---|
| | Mean ± SD | Mean ± SD | |
| Age (Years) | 58.3 ± 16.9 | 46.8 ± 16.2 | <**0.001**\* |
| Hospital Stay | 3.8 ± 3.1 | 4.0 ± 2.1 | 0.519 |
| CKD Stage | 4.3 ± 0.9 | 4.8 ± 0.6 | <**0.001**\* |
| No of Comorbidities | 0.9 ± 0.9 | 1.2 ± 0.9 | **0.013**\* |
| No of Drugs | 12.3 ± 9.3 | 2.2 ± 0.9 | <**0.001**\* |
| No of Drug Interactions | 5.6 ± 6.5 | 0.3 ± 0.5 | <**0.001**\* |

CKD: Chronic Kidney Disease; independent t-test was applied, * p-value <0.05 statistically significant

independently associated with pDDIs in a private hospital. Whereas, in the public hospital, the higher number of prescribed drugs (AOR = 2.9; p = <0.001), was an independent risk factor of pDDIs [details shown in Table 4].

Table 4 enlists the top ten frequently reported drug interacting pairs along with severity and documentation levels. The most frequently identified interacting pair in private sector hospital was furosemide–aspirin (n = 28) followed by tramadol-dimenhydrinate and rosuvastatin-clopidogrel (n = 22). Whereas, Cefoperazone-furosemide (n = 20), cefepime-furosemide (n = 16), and cefotaxime-furosemide (n = 6) were the most prevalent drug interacting pairs identified in public hospital [as shown in Table 5].

## Discussion

CKD patients, due to compromised renal function, are at a higher risk for drug-related problems, including drug-drug interactions (DDIs) [35, 36]. Healthcare professionals need to pay more attention while managing patients with chronic diseases due to the significant effects of these interactions on the patient's health and its economic burden on the healthcare system [37]. Pakistan, being a developing country, the patients receiving healthcare at hospitals exposes patients to potential risks of pDDIs and other adverse or iatrogenic effects due to overburdened, loss of follow-up and no facility available for scanning of pDDIs on spot for the patients [38]. This study compared the incidence of pDDIs between a public hospital (run by the government) and a private hospital among CKD patients.

The study found that the incidence of pDDIs was significantly higher in the private hospital (84.7%) than in the public hospital (26.6%). This result is consistent with previous studies conducted in Turkey, Nepal, Pakistan, and India that reported pDDIs rates ranging from 69.7% to 89.1% among CKD patients [15, 25, 39, 40]. The higher incidence of pDDIs in the private hospital may be due to a higher number of drugs prescribed, which increases the risk of pDDIs. These results are in line with the findings of other studies [41–43]. It is important to note that the differences in pDDIs rates among studies can be attributed to variations in study design, population characteristics, methodology, classification of interactions, definitions of pDDIs, and prescribing practices in different countries. The results of this study suggest that patients with CKD are at an increased risk for pDDIs. To minimize, prevent, or manage these interactions in a hospital setting, several evidence-based strategies have been proposed, including using computerized screening programs to identify pDDIs [44], involving clinical pharmacists in the assessment of pDDIs [45–47], utilizing structured evaluation methods [48] and evaluating relevant laboratory investigations to determine the clinical relevance of potential interactions [49, 50]. The results of this study suggest that patients with CKD are at an increased risk

**Table 4. Logistic regression analyses.**

| Variables | Private sector Hospital | | | | Public Sector Hospital | | | |
|---|---|---|---|---|---|---|---|---|
| | Univariate Analysis | | Multivariate Analysis | | Univariate Analysis | | Multivariate Analysis | |
| | OR (95% CI) | P-value | AOR (95% CI) | P-value | OR (95% CI) | P-value | AOR (95% CI) | P-value |
| **Gender** | | | | | | | | |
| Female | Reference | | Reference | | Reference | | Reference | |
| Male | 0.8 (0.3–1.9) | 0.63 | - | | 0.7 (0.3–1.4) | 0.281 | 0.5 (0.2–1.2) | 0.131 |
| **Age (Years)** | | | | | | | | |
| = <40 | Reference | | Reference | | Reference | | Reference | |
| 41–60 | 5.6 (1.6–19.9) | **0.008**\* | 6.2 (1.6–24.1) | **0.008**\* | 1.6 (0.7–3.2) | 0.224 | 1 (0.4–2.4) | 0.911 |
| >60 | 2 (0.7–5.8) | 0.182 | 1.9 (0.6–6) | 0.268 | 0.6 (0.2–1.9) | 0.392 | 0.4 (0.1–1.5) | 0.17 |
| **Hospital Stay** | | | | | | | | |
| = <2 | Reference | | Reference | | Reference | | Reference | |
| 3–4 | 2.9 (1.1–7.8) | **0.038**\* | 2 (0.6–5.9) | 0.212 | 3.2 (1–9.7) | **0.047**\* | 2.8 (0.8–9.1) | 0.08 |
| >4 | 3.2 (1–10.3) | **0.048**\* | 3 (0.8–10.3) | 0.083 | 1.8 (0.5–6.3) | 0.36 | 1.7 (0.4–6.2) | 0.427 |
| **No of Prescribed Drugs** | 1.2 (1–1.4) | **0.009**\* | 1.2 (1–1.4) | **0.027**\* | 2.6 (1.7–3.9) | **<0.001**\* | 2.9 (1.7–4.6) | **<0.001**\* |
| **Comorbidities** | | | | | | | | |
| No | Reference | | Reference | | Reference | | Reference | |
| Yes | 1.9 (0.9–4.5) | 0.108 | 4.10.8130.087 | 0.087 | 2.1 (0.8–5.1) | 0.102 | 0.5 (0.1–3.3) | 0.461 |
| **No of Comorbidities** | | | | | | | | |
| 0 | Reference | | Reference | | Reference | | Reference | |
| 1 | 1.5 (0.6–3.6) | 0.333 | 1.40.50.509 | 0.509 | 1.5 (0.5–3.8) | 0.43 | 0.8 (0.2–3.2) | 0.786 |
| = >2 | 4.6 (0.9–21.7) | 0.054 | 4.20.80.086 | 0.086 | 3.2 (1.2–8.5) | **0.018**\* | 0.6 (0.1–4.3) | 0.645 |
| **CKD Stage** | | | | | | | | |
| II | Reference | | Reference | | Reference | | | |
| III | 0.7 (0.1–6.9) | 0.756 | - | | 0.6 (0.03–14) | 0.794 | - | - |
| IV | 1.5 (0.1–15.3) | 0.754 | - | | 2.4 (0.2–32.8) | 0.512 | - | - |
| V | 0.8 (0.1–7.7) | 0.896 | - | | 1.5 (0.2–13.4) | 0.738 | - | - |
| **Comorbidities** | | | | | | | | |
| Hypertension | 2.1 (0.4–9.7) | 0.321 | - | | 1.6 (0.7–3.2) | 0.183 | 1.9 (0.6–5.8) | 0.248 |
| Diabetes Mellitus | 3.3 (0.4–26) | 0.253 | 1.5 (0.1–21.7) | 0.768 | 3.1 (1.5–6.4) | **0.002**\* | 3.4 (1.1–10.5) | 0.033 |
| Hepatitis C Virus | 1.7 (0.5–5.2) | 0.365 | - | | 0.7 (0.2–2.4) | 0.661 | - | - |
| Heart Disease | 1.1 (0.2–5.1) | 0.916 | - | | 4.1 (1.4–11.9) | **0.008**\* | 4.9 (1.3–18.8) | 0.021 |
| Urinary Tract Infection | 0.9 (0.1–7.9) | 0.922 | | | 0.7 (0.2–2.7) | 0.652 | | |
| Hepatitis B Virus | 1.4 (0.5–3.4) | 0.445 | | | 0.5 (0.1–3.8) | 0.464 | | |
| Benign Prostate Hyperplasia | 1.5 (0.1–12.2) | 0.724 | | | 2.8 (0.2–45.7) | 0.469 | | |

Multivariate logistic regression was applied, \* p-value <0.05 was statistically significant

for pDDIs and that appropriate preventive measures and interventions should be taken to minimize the risk.

Additionally, our findings revealed that CKD patients in public hospital had more comorbidities than those in private hospital. This finding aligns with a study by Gowada et al., which showed that patients in public hospitals had more comorbidities than those in private hospitals [51]. We also found that the risk of pDDIs was 6.2 times higher in patients aged 41–60 years in private hospital. Interestingly, the literature has shown that increasing patient age is independently associated with multiple comorbidities [52], as there is a mutual amplification of comorbid conditions and risks associated with CKD. However, we did not find any significant association of comorbidities with pDDIs in private hospital, while the risk of pDDIs was 3.2

**Table 5. Top ten most frequently identified interacting pairs along with severity and documentation levels.**

| Private Sector Hospital | | | | Public Sector Hospital | | | |
|---|---|---|---|---|---|---|---|
| Interacting Pairs | Severity | Documentation | n (%) | Interacting Pairs | Severity | Documentation | n (%) |
| Furosemide—Aspirin | Moderate | Fair | 28 (2.7) | Cefoperazone—Furosemide | Moderate | Fair | 20 (11.1) |
| Tramadol—Dimenhydrinate | Major | Fair | 24 (2.4) | Cefepime—Furosemide | Moderate | Fair | 16 (8.9) |
| Rosuvastatin—Clopidogrel | Moderate | Good | 22 (2.2) | Cefotaxime—Furosemide | Moderate | Fair | 6 (3.3) |
| Enoxaparin—Clopidogrel | Moderate | Fair | 17 (1.7) | Furosemide—Aspirin | Moderate | Good | 2 (1.1) |
| Moxifloxacin—Aspirin | Moderate | Poor | 14 (1.4) | Captopril—Furosemide | Moderate | Good | 1 (0.6) |
| Enoxaparin—Aspirin | Moderate | Fair | 13 (1.3) | Ciprofloxacin -Spironolactone | Major | Fair | 1 (0.6) |
| Aspirin—Clopidogrel | Moderate | Fair | 12 (1.2) | Piperacillin -Vancomycin | Moderate | Good | 1 (0.6) |
| Clopidogrel—Pantoprazole | Major | Fair | 12 (1.2) | Ramipril—Aspirin | Moderate | Fair | 1 (0.6) |
| Heparin—Clopidogrel | Moderate | Good | 12 (1.2) | Ramipril—Furosemide | Moderate | Good | 1 (0.6) |
| Clopidogrel—Omeprazole | Major | Good | 9 (0.9) | Spironolactone—Furosemide | Moderate | Fair | 1 (0.6) |

times higher in public hospital. This may be due to an overburdened nephrologist, a lack of follow-up visits, and a lack of pDDIs scanning facilities in public hospital.

Our study also showed that with each unit increase in the number of drugs, the risk of pDDIs increased by 1.2 times in private hospital compared to 2.9 times in public hospital. The higher risk of pDDIs with an increase in the number of drugs may be due to the compromised renal function of patients in public hospital, as evident from our data showing that the majority of patients had worse kidney conditions than those in private hospital.

Regarding the severity and documentation of pDDIs, we found that the majority of pDDIs were of moderate severity in private hospital (79.0%) compared to public hospital. Our findings are consistent with another study reporting 75.1% of pDDIs of moderate severity in CKD patients [15], while another study reported 20% major, 57% moderate, and 23% minor pDDIs in CKD patients [53]. In Pakistan, another study reported 60.8% moderate, 41.1% minor, and 27.8% major pDDIs [25]. Regarding the documentation of pDDIs, the majority of pDDIs were of fair documentation grade in both public and private hospital, which is consistent with other studies [25, 54].

Our study found that CKD patients are at risk of pDDIs, which can have adverse clinical consequences. Therefore, it is essential for healthcare professionals to identify the specific type of pDDIs and develop therapeutic guidelines to prevent associated risks and ensure effective clinical management of these interactions. By improving their knowledge and understanding of pDDIs, physicians can help minimize the occurrence of adverse events and enhance the quality of care for CKD patients.

The severity of pDDIs is always clinically significant. Therefore, it is crucial to develop a comprehensive list of the most commonly observed and clinically important interactions. This list can then be utilized by physicians and pharmacists to establish therapeutic guidelines and proactively and promptly identify pDDIs. With a better understanding of pDDIs, physicians can contribute to reducing the occurrence of adverse events associated with medication use, adjust treatment plans for patients at higher risk of pDDIs, improve the overall quality of care, and mitigate any medico-legal concerns.

## Strengths & limitations

This study is the first of its kind in Pakistan to compare the patterns of pDDIs in CKD patients between private and public hospitals. However, there are a few limitations to consider. The study only included one private and one public hospital, and the inclusion of other diseases

and multiple hospitals could provide a more comprehensive understanding of pDDIs and rational prescribing practice among different healthcare settings.

## Conclusion

The study highlighted a high incidence of pDDIs in CKD patients receiving care in private hospitals, with most of these interactions being of moderate severity. Furthermore, a significant number of patients also experienced major pDDIs. The risk of experiencing pDDIs was found to be higher in older patients and those taking a higher number of drugs. To enhance patient safety and improve treatment outcomes, the study recommends implementing various strategies such as involving pharmacists in assessment of pDDIs to alleviate the workload of nephrologists, utilizing software-based screening for pDDIs, providing comprehensive patient education and counseling, and establishing regular monitoring and follow-up procedures. By adopting these strategies, healthcare professionals can effectively address the challenges posed by pDDIs and optimize the care provided to CKD patients.

## Supporting information

**S1 File. STROBE statement.**
(DOCX)

**S2 File. Inclusivity in global research.**
(DOCX)

## Author Contributions

**Conceptualization:** Roheena Zafar, Inayat Ur Rehman.

**Data curation:** Roheena Zafar, Yasar Shah, Long Chiau Ming.

**Formal analysis:** Roheena Zafar, Inayat Ur Rehman, Long Chiau Ming, Khang Wen Goh.

**Funding acquisition:** Inayat Ur Rehman, Long Chiau Ming, Hui Poh Goh.

**Investigation:** Roheena Zafar, Inayat Ur Rehman, Yasar Shah, Hui Poh Goh.

**Methodology:** Roheena Zafar, Inayat Ur Rehman, Long Chiau Ming.

**Project administration:** Inayat Ur Rehman, Yasar Shah, Khang Wen Goh.

**Supervision:** Inayat Ur Rehman, Yasar Shah.

**Validation:** Roheena Zafar.

**Visualization:** Hui Poh Goh.

**Writing – original draft:** Roheena Zafar.

**Writing – review & editing:** Inayat Ur Rehman, Yasar Shah, Long Chiau Ming, Hui Poh Goh, Khang Wen Goh.

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
