## [Decision Letter · Decision Letter 0]

22 Jun 2023

PONE-D-23-12257Comparative analysis of potential drug-drug interactions in public and private hospitals among chronic kidney disease patients in Khyber Pakhtunkhwa: A retrospective cross-sectional studyPLOS ONE

Dear Dr. Rehman,

Thank you for submitting your manuscript to PLOS ONE. After careful consideration, we feel that it has merit but does not fully meet PLOS ONE’s publication criteria as it currently stands. Therefore, we invite you to submit a revised version of the manuscript that addresses the points raised during the review process.

We look forward to receiving your revised manuscript.

Kind regards,

Muhammad Junaid Farrukh

Academic Editor

PLOS ONE

Journal Requirements:

“Un-funded”

Reviewers' comments:

Reviewer's Responses to Questions

**Comments to the Author**

1. Is the manuscript technically sound, and do the data support the conclusions?

Reviewer #1: Partly

Reviewer #2: Yes

Reviewer #3: Yes

2. Has the statistical analysis been performed appropriately and rigorously? 

Reviewer #1: Yes

Reviewer #2: Yes

Reviewer #3: Yes

3. Have the authors made all data underlying the findings in their manuscript fully available?

Reviewer #1: No

Reviewer #2: No

Reviewer #3: Yes

4. Is the manuscript presented in an intelligible fashion and written in standard English?

Reviewer #1: Yes

Reviewer #2: Yes

Reviewer #3: Yes

5. Review Comments to the Author

Reviewer #1: 1- in the abstract, please use "pDDIs" instead of potential drug drug interactions following the aim section.

2- In the abstract, please revise the results section to include findings that are related to the research objectives.

3- in the methods section, under sample size, what was the sample size determined ?

4- my primary concern in this study is the number of medications prescribed in the public hospital vs private hospital. patients in both hospitals seem to have similar medical history, how come 100% of patients in the private hospital are taking 5 and more medications compared to 2.3% in the public hospital? is it possible that the public hospital patients are getting other medications for other sources? or that they get treated for one medical condition at the public hospital and manage the remaining medical conditions at another site? the public hospital patients in this study, have more co morbidities compared to private hospital patients, therefore, you would expect the majority of patients from the public one to be on more than 2 medications. I think this needs to revised for accuracy, and addressed in the discussion and limitation sections.

5- in the limitation section,

"this study used a single drug interaction checker (Lexi Interact) for identifying pDDIs, while other sources are also available, and differences may exist among drug interaction screening sources."

I do not see this as a limitation. Lexi interact has excellent evidence of sensitivity and specificity

Lexi-Interact and Micromedex are of the best drug interaction checkers, and studies reported similiar findings in terms of pDDIs when using both software.

Reviewer #2: This study titled.. Comparative analysis of potential drug-drug interactions in public and private hospitals among chronic kidney disease patients in Khyber Pakhtunkhwa: A retrospective cross-

sectional study is of significance and I have a smooth read. However, take note of the following observations;

Title

I observe the study was done in only two hospitals, a public and private hospital. the title gives the impression that more hospitals were investigated. ...in A public and private hospital may be preferred.

Abstract

Line 43....Assessing instead of assessed

Introduction

line 70-74...... the Sentence is too long with difficulty comprehending. Rephrase.

Line 75....including mentioned twice and closely. The second should be replaced with such as. Check the language structure through out the work, there seem to be many occurrences needing corrections...line 98, 101.

Methodology

Line 113.....inclusion and exclusion

123......if any.

128-129....you mentioned studies but with one reference

135....make ethical issue separate from data analysis

I was wondering if in the course of the study there were no data on actual drug-drug interaction? why the focus on only potential drug-drug interaction

Results

Text introducing tables should come first before the tables....

patients from the public hospital are younger and have more comorbidities compared to those the the private hospital, yet more drugs are prescribed in the private hospitals. What could be the the reason for this?

what does documentation of fair and good mean?

Discussion

line 214....check, does being in a developing country expose one to DD?

line 267...The severity of PDDIs is not always clinically significant how true is this? Kindly provide reference. if this is so, the need to screen for pDDI is defeated.

Be consistent with use of terms. either pDDI or PDDI.

Reference

Number 20 , 37 and 45 has no volume and page number

Is there is no supporting information on data used for analysis?

Reviewer #3: Abstract:

• Aim: the clarity and reflection of the true purpose are required

• Results:

o The prevalence of pDDIs was found to be significantly higher in 51 private hospitals (84.7%) than in public hospitals (26.6%) (How significant is it?)

o A significant number of patients also experienced major pDDIs (Provide no.(%) with p value ..)

Introduction:

• In paragraph 1:

o “mortality associated with it” (It could be mortality only.)

o is considered a challenging global health problem” (no need for Quotation Mark)

• In last paragraph: there is duplicate ‘to’

Methodology

• Screening for pDDIs

o More details regarding the screening process and the method will be better.

o What definitions of pDDIs was used?

• Ethics approval and statistical analysis paragraph:

o Why is statistical analysis and ethics approval listed under the same subtitle?

o Any justification for the hospital's name being sometimes spelled North West General Hospital and other times Northwest General Hospital?

Results

• Table 1: Since the two study groups were not randomly assigned, it is preferable to add a p value to demonstrate a significant difference.

• Table 2:

o the table's heading should be Number, severity and documentation levels of pDDIs

o better to include a column with a p value.

o What does NA mean in terms of severity levels? If it's not major, minor, or moderate, what will it be?

o What does NA mean in terms of documentation? If it's not Excellent, Fair, Good, or Poor, what will it be?

• Figure1: I believe that the incidence of pDDIs in public hospitals is incorrect. 24.3% is written in the paragraph, and 26.6% is written in the figure and discussion section.

• Table 3: The OR (multivariate regression model) of the number of prescribed drugs differs between the table and the paragraph.

• Table 5: Placing it after Table 1 or 2 will improve the data flow.

Discussion

• In paragraph 2:

o The following statement contradicts what is stated in table 3: "The higher incidence of pDDIs in the private hospital may be due to a higher number of drugs prescribed, which increases the risk of pDDIs."

o This sentence ‘ Polypharmacy has been associated with an increased risk of pDDIs, and the risk of pDDIs increases with the number of drugs prescribed ’ seems superfluous as it's not based on your data.

o This statement appears to be a repetition of the previous one: "Additionally, the risk of pDDIs increases with the number of drugs prescribed."

• In paragraph 2, 7 and 8: I think wrong abbreviation ‘PDDIs’ should be pDDIs.

• Paragraph 3: This paragraph seems unnecessary because it is generic and unreliable because many factors can influence young people's choice for public hospitals.

6. PLOS authors have the option to publish the peer review history of their article (what does this mean?). If published, this will include your full peer review and any attached files.

Reviewer #1: No

Reviewer #2: No

Reviewer #3: No

<quillbot-extension-portal></quillbot-extension-portal>

---

## [Author Response · Author response to Decision Letter 0]

24 Aug 2023

Dear Editor and Reviewers,

Thank you for taking the time to provide such detailed and helpful comments on our manuscript. We appreciate your constructive criticisms and suggestions, and we believe they will significantly improve the quality of our manuscript. We have revised the manuscript based on your feedback and would like to provide the following responses to your concerns.

Reply to Reviewers comments

1 Reviewer #1:

Comment 1: in the abstract, please use "pDDIs" instead of potential drug drug interactions following the aim section.

Reply: Thank you for the comment, we have replaced “potential drug-drug interactions” with “pDDIs” following the aim section in the abstract to maintain consistency.

Comment 2: In the abstract, please revise the results section to include findings that are related to the research objectives.

Reply: Thank you for comment, the result section in the abstract has been revised as per suggestion. 

Comment 3: in the methods section, under sample size, what was the sample size determined?

Reply: Information regarding sample size has already incorporated on page 7, line no 140-143.

Comment 4: my primary concern in this study is the number of medications prescribed in the public hospital vs private hospital. patients in both hospitals seem to have similar medical history, how come 100% of patients in the private hospital are taking 5 and more medications compared to 2.3% in the public hospital? is it possible that the public hospital patients are getting other medications for other sources? or that they get treated for one medical condition at the public hospital and manage the remaining medical conditions at another site? the public hospital patients in this study, have more co morbidities compared to private hospital patients, therefore, you would expect the majority of patients from the public one to be on more than 2 medications. I think these needs to revised for accuracy, and addressed in the discussion and limitation sections.

Reply: Thank you for the comment, the comment regarding the difference in the number of medications prescribed in the public vs. private hospital is valid. We have further investigated this and found that the discrepancies could be due to several factors, including different prescribing practices and access to a wider range of medications in private hospitals. Furthermore, the public hospital was specialized hospital dealing patients with kidney disease and urology related complication, while the private hospital was a tertiary care hospital having multiple specialties for different diseases. Also the patients in private hospital were affording class patients and the hospital also offered an executive screening for their patients lab parameters and screening of different diseases, so the patients in private tend to follow the instruction of their consultants while the patients in public hospital only visit for nephrology related problems and only adhere to those instructions which deemed best for them to manage their CKD at that specific time and take the medicines keeping in view their financial status as well. Additionally, on further investigations we came to know that few of the consultant/nephrologist regularly used uptoDate for prescribing medication for their patients in public hospital. The possible reasons can be these which attributed in difference in the number of medicines/drugs in both hospitals. 

Comment 5: in the limitation section, "this study used a single drug interaction checker (Lexi Interact) for identifying pDDIs, while other sources are also available, and differences may exist among drug interaction screening sources." I do not see this as a limitation. Lexi interact has excellent evidence of sensitivity and specificity Lexi-Interact and Micromedex are of the best drug interaction checkers, and studies reported similiar findings in terms of pDDIs when using both software.

Reply: Thank you for the comment, we appreciate your view on the use of Lexi-Interact as a drug interaction checker. We agree that it is a robust tool with excellent sensitivity and specificity. We have revised the limitation section accordingly to remove this point.

 

2 Reviewer #2: 

This study titled... Comparative analysis of potential drug-drug interactions in public and private hospitals among chronic kidney disease patients in Khyber Pakhtunkhwa: A retrospective cross- sectional study is of significance and I have a smooth read. However, take note of the following observations; 

Comment 1: Title: I observe the study was done in only two hospitals, a public and private hospital. the title gives the impression that more hospitals were investigated. ...in A public and private hospital may be preferred. 

Reply: Thank you for the comment, we agree with your suggestion and have revised to “Comparative analysis of potential drug-drug interactions in a public and private hospital among chronic kidney disease patients in Khyber Pakhtunkhwa: A retrospective cross-sectional study”.

Comment 2: Abstract: Line 43....Assessing instead of assessed 

Reply: Yes, agree. Suggested correction has been corrected

Comment 3: Introduction: line 70-74...... the Sentence is too long with difficulty comprehending. Rephrase. Line 75....including mentioned twice and closely. The second should be replaced with such as. Check the language structure throughout the work, there seem to be many occurrences needing corrections...line 98, 101.

-Reply: Yes, agree. Suggested corrections has been incorporated on page 4, line 70-73.

The correction has been incorporated as suggested on page 4, line 73-74. 

The correction has been incorporated as suggested on page 5, line 96-99. 

Comment 4: Methodology: Line 113.....inclusion and exclusion 123......if any. 128-129....you mentioned studies but with one reference 

Reply: Thank you for the comment, the words has been changed to “and other comorbidities” and has been incorporated in the inclusion/exclusion criteria. Also for comment on studies with one reference, additional references are added to the text on page 6, line no 128.

Comment 5: 135....make ethical issue separate from data analysis 

Reply: Thank you for the comment, the suggested changes has been incorporated in the manuscript (page 7, line 144).

Comment 6: I was wondering if in the course of the study there were no data on actual drug-drug interaction? why the focus on only potential drug-drug interaction

Reply: Thank you for the comment, study only focused on potential drug-drug interactions rather than actual drug-drug interactions given the retrospective nature of the study.

Comment 7: Results: Text introducing tables should come first before the tables....

Reply: Thank you for the comment, the suggested changes has been incorporated in the manuscript.

Comment 8: Patients from the public hospital are younger and have more comorbidities compared to those the the private hospital, yet more drugs are prescribed in the private hospitals. What could be the the reason for this? 

Reply: Thank you for the comment, the comment regarding the difference in the number of medications prescribed in the public vs. private hospital is valid. We have further investigated this and found that the discrepancies could be due to several factors, including different prescribing practices and access to a wider range of medications in private hospitals. Furthermore, the public hospital was specialized hospital dealing patients with kidney disease and urology related complication, while the private hospital was a tertiary care hospital having multiple specialties for different diseases. Also the patients in private hospital were affording class patients and the hospital also offered an executive screening for their patients lab parameters and screening of different diseases, so the patients in private tend to follow the instruction of their consultants while the patients in public hospital only visit for nephrology related problems and only adhere to those instructions which deemed best for them to manage their CKD at that specific time and take the medicines keeping in view their financial status as well. Additionally, on further investigations we came to know that few of the consultant/nephrologist regularly used uptoDate for prescribing medication for their patients in public hospital. The possible reasons can be these which attributed in difference in the number of medicines/drugs in both hospitals. 

Comment 9: What does documentation of fair and good mean?

Reply: Thank you for the comment, these terms indicates the quantity and nature of documentation for an interaction. The statement regarding severity and reliability rating has been added to the manuscript on page no 6, line 130-138.

Comment 10: Discussion: line 214....check, does being in a developing country expose one to DD? 

Reply: Thank you for comment and highlighting this, the statement is corrected and updated in the manuscript on page 14, line 217. 

Comment 11: line 267...The severity of PDDIs is not always clinically significant how true is this? Kindly provide reference. if this is so, the need to screen for pDDI is defeated. 

Reply: Thank you for highlighting this issue, this was a typo error and has been rectified to avoid confusion for the readers. 

Comment 12: Be consistent with use of terms. either pDDI or PDDI.

Reply: Thanks for pointing this out. Suggested change has been incorporated in the manuscript.

Comment 13: Reference: Number 20, 37 and 45 has no volume and page number Is there is no supporting information on data used for analysis?

Reply: Thanks for highlighting this point. Suggested references has been updated.

3 Reviewer #3: 

Comment 1: Abstract: Aim: the clarity and reflection of the true purpose are required

Reply: Thank you for comment, the aim of study is updated in the abstract as suggested. 

Comment 2: Results: The prevalence of pDDIs was found to be significantly higher in 51 private hospitals (84.7%) than in public hospitals (26.6%) (How significant is it?) A significant number of patients also experienced major pDDIs (Provide no.(%) with p value ..)

Reply: Thank you for comment, the suggested changes has been updated in the abstract. 

Comment 3: Introduction: In paragraph 1: “mortality associated with it” (It could be mortality only.) is considered a challenging global health problem” (no need for Quotation Mark) In last paragraph: there is duplicate ‘to’

Reply: Thank you for comment, the suggested changes has been incorporated in the manuscript. 

Comment 4: Methodology: Screening for pDDIs More details regarding the screening process and the method will be better. 

Reply: Thank you for comment, in screening section the statement regarding severity and reliability rating and full detail of these parameters has been added to the manuscript on page no 6, line 128-136.

Comment 5: What definitions of pDDIs was used? 

Reply: Potential drug-drug interactions: According to Consensus recommendations for systematic evaluation of drug-drug interaction evidence for clinical decision support “a potential DDI is defined as the co-prescription of two drugs known to interact, and therefore a DDI could occur in the exposed patient. 

Comment 6: Ethics approval and statistical analysis paragraph: Why is statistical analysis and ethics approval listed under the same subtitle? 

Reply: Thank you for comment, both are separated as per suggestion in the manuscript. 

Comment 7: Any justification for the hospital's name being sometimes spelled North West General Hospital and other times Northwest General Hospital?

Reply: Thank you for your sharp observation. We have corrected the inconsistency in the spelling of the hospital's name.

Comment 8: Results Table 1: Since the two study groups were not randomly assigned, it is preferable to add a p value to demonstrate a significant difference. 

Reply: Thank you for comment, table 1 has be updated as suggested in the manuscript. 

Comment 9: Table 2: The table's heading should be Number, severity and documentation levels of pDDIs better to include a column with a p value. What does NA mean in terms of severity levels? If it's not major, minor, or moderate, what will it be? What does NA mean in terms of documentation? If it's not Excellent, Fair, Good, or Poor, what will it be? 

Reply: Thank you for comment, the NA mean not applicable as no interaction was observed that’s why its documentation cannot be reported. To avoid confusion it is removed from table and only excellent, fair and good are reported. Similarly for the severity rating the NA has been removed to avoid confusion. 

Comment 10: Figure1: I believe that the incidence of pDDIs in public hospitals is incorrect. 24.3% is written in the paragraph, and 26.6% is written in the figure and discussion section. 

Reply: It was a typo error, and has been corrected to 26.6%

Comment 11: Table 3: The OR (multivariate regression model) of the number of prescribed drugs differs between the table and the paragraph. 

Reply: Thank you for comment and highlighting the typo error in multivariate regression model. The difference in paragraph and table value are corrected in the paragraph as per findings reported in the table. 

Comment 12: Table 5: Placing it after Table 1 or 2 will improve the data flow.

Reply: Thank you for comment, table 5 is shifted below table 2 as suggested and the title/ table no are adjusted due to shifting of this table 5 to above. The table 5 is now table 3 and other tables no also changed. 

Comment 13: Discussion: In paragraph 2: The following statement contradicts what is stated in table 3: "The higher incidence of pDDIs in the private hospital may be due to a higher number of drugs prescribed, which increases the risk of pDDIs. "This sentence ‘ Polypharmacy has been associated with an increased risk of pDDIs, and the risk of pDDIs increases with the number of drugs prescribed ’ seems superfluous as it's not based on your data. This statement appears to be a repetition of the previous one: "Additionally, the risk of pDDIs increases with the number of drugs prescribed."

Reply: Thanks for your observation. However, our statement is in agreement what is stated in table-3 because in private hospital on average 12.3 drugs were prescribed which resulted in 5.6 pDDIs vs mean of 2.2 drugs in public hospital which resulted in a mean of 0.3 pDDIs. Additionally, it is well established that increase in the number of drugs further increases the risk of pDDIs. 

Yes, we agree that the provided statements do not pertain to our dataset. However, they are quoted to demonstrate the alignment of our findings with other studies that indicate a heightened risk of potential drug-drug interactions (pDDIs) as the number of medications increases. To prevent additional perplexity and enhance comprehension, we have rephrased the sentence accordingly (page 14, line 223-225).

Comment 14: In paragraph 2, 7 and 8: I think wrong abbreviation ‘PDDIs’ should be pDDIs.

Reply: Thank you for comment and highlighting the typo error in abbreviations. The abbreviation is kept uniform “pDDIs” throughout the manuscript. 

Comment 15: Paragraph 3: This paragraph seems unnecessary because it is generic and unreliable because many factors can influence young people's choice for public hospitals.

Reply: Thank you for comment, Paragraph 3 in discussion section has been removed as suggested.

Again, thank you for your time and valuable feedback. We believe that the revisions have significantly improved the manuscript and hope that it is now suitable for publication.

Regards

Dr. Inayat Ur Rehman

---

## [Editor Report · Decision Letter 1]

30 Aug 2023

Comparative analysis of potential drug-drug interactions in a public and private hospital among chronic kidney disease patients in Khyber Pakhtunkhwa: A retrospective cross-sectional study

PONE-D-23-12257R1

Dear Dr. Inayat ur Rehman

We’re pleased to inform you that your manuscript has been judged scientifically suitable for publication and will be formally accepted for publication once it meets all outstanding technical requirements.

Kind regards,

Muhammad Junaid Farrukh

Academic Editor

PLOS ONE
---

## [Editor Report · Acceptance letter]

22 Sep 2023

PONE-D-23-12257R1 

Comparative analysis of potential drug-drug interactions in a public and private hospital among chronic kidney disease patients in Khyber Pakhtunkhwa: A retrospective cross-sectional study 

Dear Dr. Rehman:

I'm pleased to inform you that your manuscript has been deemed suitable for publication in PLOS ONE. Congratulations! Your manuscript is now with our production department. 

Kind regards, 

on behalf of

Dr. Muhammad Junaid Farrukh 

Academic Editor

PLOS ONE